# Modulation of DNA Damage Response by SAM and HD Domain Containing Deoxynucleoside Triphosphate Triphosphohydrolase (SAMHD1) Determines Prognosis and Treatment Efficacy in Different Solid Tumor Types

**DOI:** 10.3390/cancers14030641

**Published:** 2022-01-27

**Authors:** Eudald Felip, Lucía Gutiérrez-Chamorro, Maica Gómez, Edurne Garcia-Vidal, Margarita Romeo, Teresa Morán, Laura Layos, Laia Pérez-Roca, Eva Riveira-Muñoz, Bonaventura Clotet, Pedro Luis Fernandez, Ricard Mesía, Anna Martínez-Cardús, Ester Ballana, Mireia Margelí

**Affiliations:** 1AIDS Research Institute-IrsiCaixa, IGTP (Health Research Institute Germans Trias i Pujol), Hospital Germans Trias i Pujol, Universitat Autònoma de Barcelona, 08916 Badalona, Spain; efelip@iconcologia.net (E.F.); lgutierrez@irsicaixa.es (L.G.-C.); egvidal@irsicaixa.es (E.G.-V.); eriveira@irsicaixa.es (E.R.-M.); bclotet@irsicaixa.es (B.C.); 2Medical Oncology Department, Catalan Institute of Oncology-Badalona, Hospital Germans Trias i Pujol (HGTiP), 08916 Badalona, Spain; mromeo@iconcologia.net (M.R.); mmoran@iconcologia.net (T.M.); llayos@iconcologia.net (L.L.); rmesia@iconcologia.net (R.M.); amartinezc@igtp.cat (A.M.-C.); 3(B-ARGO) Badalona Applied Research Group in Oncology, (IGTP), Institut d’Investigació en Ciències de la Salut Germans Trias i Pujol, Departament de Medicina, Universitat Autònoma de Barcelona, 08916 Badalona, Spain; 4Department of Pathology, IGTP (Health Research Institute Germans Trias i Pujol), Hospital Germans Trias i Pujol, Universitat Autònoma de Barcelona, 08916 Badalona, Spain; maicameister@gmail.com (M.G.); plfernandez.germanstrias@gencat.cat (P.L.F.); 5Banc de Tumors, IGTP (Health Research Institute Germans Trias i Pujol), Hospital Germans Trias i Pujol, Universitat Autònoma de Barcelona, 08916 Badalona, Spain; laiaproca@gmail.com

**Keywords:** SAMHD1, NSCLC, breast cancer, ovarian cancer, solid tumors

## Abstract

**Simple Summary:**

Ample evidence exists connecting dNTP pool dysregulation to cancer and genomic instability. SAMHD1, the unique dNTP triphosphohydrolase described in humans, has been proposed to play a key role in hematological cancers, although its value in advanced solid tumors has not yet been explored. Moreover, it has been proposed that SAMHD1 might fulfill the requirements of a driver gene in tumor development or might promote a so-called mutator phenotype. Here, we show that low SAMHD1 expression tumors show better prognosis in breast, ovarian, and lung advanced cancer. Moreover, low SAMHD1 expression is a positive predictive factor in lung and ovarian cancer treated with platin derivates and/or antimetabolites. In vitro evaluation of its underlying mechanism revealed that SAMHD1 KO cells presented enhanced susceptibility to DNA damage and subsequent induction of apoptosis of tumor cells. Our results provide strong evidence of the clinical importance of SAMHD1, becoming an interesting target for the development of personalized cancer treatments.

**Abstract:**

SAMHD1 is a deoxynucleotide triphosphate (dNTP) triphosphohydrolase with important roles in the control of cell proliferation and apoptosis, either through the regulation of intracellular dNTPs levels or the modulation of the DNA damage response. However, SAMHD1′s role in cancer evolution is still unknown. We performed the first in-depth study of SAMHD1′s role in advanced solid tumors, by analyzing samples of 128 patients treated with chemotherapy agents based on platinum derivatives and/or antimetabolites, developing novel in vitro knock-out models to explore the mechanisms driving SAMHD1 function in cancer. Low (or no) expression of SAMHD1 was associated with a positive prognosis in breast, ovarian, and non-small cell lung cancer (NSCLC) cancer patients. A predictive value was associated with low-SAMHD1 expression in NSCLC and ovarian patients treated with antimetabolites in combination with platinum derivatives. In vitro, SAMHD1 knock-out cells showed increased γ-H2AX and apoptosis, suggesting that SAMHD1 depletion induces DNA damage leading to cell death. In vitro treatment with platinum-derived drugs significantly enhanced γ-H2AX and apoptotic markers expression in knock-out cells, indicating a synergic effect of SAMHD1 depletion and platinum-based treatment. SAMHD1 expression represents a new strong prognostic and predictive biomarker in solid tumors and, thus, modulation of the SAMHD1 function may constitute a promising target for the improvement of cancer therapy.

## 1. Introduction

Deoxynucleotide triphosphates (dNTPs) are the raw materials for DNA replication and repair, rendering them indispensable components for cell division and the maintenance of genomic stability [1]. Sterile alpha motif and histidine/aspartic acid domain-containing protein 1 (SAMHD1), the only dNTP triphosphohydrolase in eukaryotes, plays a key role in cell metabolism and, thus, has been linked to several pathological processes. SAMHD1 is well known for its role as a viral restriction factor, particularly in nondividing cells by limiting dNTPs required for viral transcription and replication [2]. SAMHD1 is a dNTPase that catalyzes the conversion of dNTP to deoxyribonucleoside (dN) and triphosphate [3]. It has been recognized that SAMHD1 may play an important role in the regulation of cellular dNTP levels, which are critical to the fidelity of DNA synthesis and the stability of the genome [4]. Mutations in SAMHD1 also cause Aicardi Goutières syndrome (AGS), a rare congenital neurodegenerative autoimmune disorder, characterized by a dysregulated interferon (IFN) signaling [5]. Due to SAMHD1 central role in cellular dNTP metabolism, its involvement in cancer development has been extensively investigated, albeit its specific role is somewhat controversial. On one hand, somatic mutations in SAMHD1 have been linked to several human cancers, being recurrently mutated in chronic lymphocytic leukemia (CLL) [6,7], frequently mutated in colorectal cancer [8], and other mutations have been found in a number of different cancers, including myeloma [9], breast cancer [10], lung carcinoma [11], pancreatic cancer [12], and glioblastoma [13]; overall suggesting that SAMHD1 may function as a tumor suppressor. This has been related to the fact that the overexpression of SAMHD1 is associated with a reduction in cell proliferation, probably due to the depletion of the dNTPs necessary for the correct replication genomic DNA [14]. In contrast, increased SAMHD1 mRNA expression has been associated with metastasis in colon cancer [15], and higher SAMHD1 serum levels have been associated with NSCLC cancer progression [16]; therefore, suggesting that low SAMHD1 expression might also represent a favorable prognostic factor in certain cancers [17].

Furthermore, SAMHD1 has been shown to modulate in vitro efficacy of several antinucleoside metabolite drugs used in the treatment of viral infections or cancer, either improving its action by depleting the intracellular pool of dNTP competitors [18,19] or limiting its action by directly using the triphosphate compounds as substrates, as in the case of cytarabine (Ara-C), which is the standard treatment for acute myeloid leukemia (AML) [20,21]. Therefore, SAMHD1 inhibition is considered a promising strategy to overcome tumor resistance and SAMHD1 expression has been proposed as a potential biomarker for the stratification of patients with cancer diagnosis that have to be treated with antimetabolites [22].

From a functional point of view, SAMHD1, through its dNTP hydrolase enzymatic activity, plays a key role in the maintenance of homeostasis of cellular dNTP pools being essential for preserving genome integrity [23]. It has been reported that dNTP pool imbalance caused by SAMHD1 deficiency may lead to DNA damage, accompanied by the activation of IFN signaling [24]. In addition, high intracellular levels of dNTPs increase the mutation rate during cellular DNA replication, which is an important molecular mechanism of tumorigenesis [25]. Moreover, imbalanced dNTP levels due to SAMHD1 function dysregulation might be associated with the rate of replication fork formation under DNA replication stress, leading to gene mutations, genomic instability, and cancer development [26,27]. Therefore, SAMHD1 is considered a key regulator involved in the maintenance of the dNTP pool and genome homeostasis [28].

Consistent with these findings, changes in SAMHD1 cause DNA damage hypersensitivity; however, somewhat paradoxical to its role in dNTP pool regulation, overexpressed SAMHD1 also localizes to DNA damage sites [6]. Several studies have demonstrated that SAMHD1 participates in the DNA damage response (DDR) process, independently of its canonical dNTP hydrolase activity, indicating a novel association between SAMHD1 and DDR process, which suggest that SAMHD1 may also contribute to anticancer therapy affecting cell proliferation and survival following DNA damage induction [29]. However, the precise molecular mechanisms underlying these observations are unclear at present, and SAMHD1 influence in cancer onset, progression and/or treatment efficacy remain largely unknown. 

Similarly, no data exists on the clinical value of SAMHD1 function in cancer onset and/or progression, as well as the relative contribution of its dNTPase function that controls intracellular dNTPs pool or its role in the repair of double strand breaks (DSB) in response to DNA damage that promote genome integrity. Here, we evaluated the prognostic and predictive value of SAMHD1 expression in different solid tumors treated with platinum derivatives and/or antimetabolites. Moreover, we developed novel in vitro models to explore the mechanisms driving SAMHD1 function in cancer development and treatment response.

## 2. Materials and Methods

### 2.1. Patients and Samples

A total of 128 patients with solid tumors treated with different chemotherapy agents at the Medical Oncology Service, ICO-Badalona, from 2012 to 2018, participated in the study, including the following tumor types: 46 samples of patients with advanced breast cancer treated with capecitabine, 22 samples of advanced ovarian cancer patients treated with cisplatin or carboplatin in combination with gemcitabine or gemcitabine in monotherapy, 16 samples from NSCLC patients treated with cisplatin or carboplatin in combination with gemcitabine or pemetrexed, or with gemcitabine in monotherapy; 14 samples of advanced pancreatic cancer patients treated with nab-paclitaxel in combination with gemcitabine, and 30 samples of locally advanced rectal cancer treated with radiotherapy in combination with capecitabine (Table 1 and Appendix A). The study was conducted under the ethics principles of the Declaration of Helsinki and approved by the Research and Ethics Committee of Hospital Germans Trias i Pujol. Samples were obtained from the Biobank of the Institut d’Investigació Germans Trias i Pujol. All patients provided written informed consent. Treatment regimens are described in Appendix A.

*Cohort variables.* The following demographic, clinical and biological data and treatment algorithms were collected for all study participants: sex, date of birth, date of cancer diagnosis, date of metastasis, number of lines of therapy for advanced disease previous to cohort therapy, date of starting cohort therapy, date of end cohort therapy, cause of need cohort therapy, evaluation of overall response rate (ORR), date of last follow-up, status at last follow-up, and toxicity parameters according to National Cancer Institute Common Toxicity Criteria [30]. Tumor stage was classified according to TNM classification of the Union International Cancer Control [31]. Treatments were obtained from review of medical records. 

### 2.2. Construction of Tissue Microarrays (TMA) and Immunohistochemical Methods

For breast, ovarian, NSCLC and pancreas tumors, three different areas/tumor were selected and included into the TMA (cylinders of 0.6 mm in diameter of each block of paraffin-embedded tissue) using a TMA workstation MTA-1 (Beecher instruments, Sun Prairie, WI, USA). Then, TMAs were cut in 5-micron sections for analysis by immunohistochemistry of SAMHD1 expression (1:200, polyclonal rabbit anti-SAMHD1 antibody, cat. no. 12586-1-AP, Proteintech, Rosemont, IL, USA), using an automated detection system (ultraView, Ventana 9 after antigen retrieval). The specificity of the polyclonal antibody was validated by western blot analysis in cell lines (Appendix A) and by immunohistochemistry using paraffin-embedded tissue (Appendix A). Evaluation of the immunostained slides was performed blinded to any clinical data by experienced pathologists, reporting the level of SAMHD1 expression as the percentage of positive tumor cells. Independent triplicate evaluations were performed for each tumor. The percentage of SAMHD1 stained cells was arbitrarily defined as positive or negative, being SAMHD1 positive cases those with cellular positivity ≥25%, as previously performed in AML patients [32]. All immunohistochemical analyses were performed in the histopathological unit of Hospital Germans Trias I Pujol. 

Rectal tumors were evaluated in paraffin-embedded tissue slides following the same procedure described above but from whole tissue slides.

### 2.3. Cells Lines and SAMHD1 Knock-Out Generation

Human T47D cells were obtained from Sigma-Aldrich-ECACC (European Collection of Authenticated cell cultures, 85102201-1VL) and grown in Dulbecco’s Modified Eagle Medium (DMEM, Thermo Fisher, Madrid, Spain) supplemented with 10% of heat-inactivated fetal bovine serum (FBS, Thermo Fisher, Madrid, Spain) and antibiotics (100 U/mL penicillin, 100 μg/mL streptomycin (Life Technologies, Madrid, Spain) and maintained at 37 °C in a 5% CO_2_ incubator.

For the generation of knock-out (KO) cells, T47D cells were transfected with a plasmid expressing a CRISPR-Cas9 construct designed to disrupt the sequence corresponding to exon 5 of SAMHD1 gene that encodes for HD domain (CRISPR-SAMHD1), as described previously [33]. Briefly, 1.5 × 10^5^ cells were seeded in 24 well plates. After overnight culture, 0.5 μg of CRISPR-SAMHD1 plasmid were mixed with lipofectamine 2000 reagent (Invitrogen, Barcelona, Spain) in serum-free medium Opti-MEM (Invitrogen) and then added to previously washed cells. Media was replaced by fresh DMEM, four hours after transfection, and left in the incubator for 3 days. Cells were then treated with puromycin (1 µg/mL) for 24 h. After puromycin selection, single cell clones were obtained by limiting dilution in 96-well plates. Once grown, SAMHD1 KO clones were identified by testing for the presence of SAMHD1 protein expression by western blot. Control cells (WT) were generated in parallel and used in all experiments.

OVCAR-3 cells were obtained from American Type Culture Collection (ATCC) and grown in RPMI 1640 medium supplemented with 10% heat-inactivated fetal bovine serum (FBS; Thermo Fisher Scientific, Waltham, MA, USA), 100 U/mL penicillin, 100 μg/mL streptomycin (Life Technologies), and insulin solution human (0.01 mg/mL) (Sigma-Aldrich, Saint Louis, MO, USA) and maintained at 37 °C in a 5% CO_2_ incubator.

For SAMHD1 knock-down in OVCAR-3 cells, siRNAs targeting SAMHD1 gene (siSAMHD1) and non-targeting control (siNT) (siGENOME SMARTpool; Dharmacon, Cultek) were transfected following standard procedures. In brief, 100 nM of siRNA were mixed with Lipofectamine 3000 reagent (Thermo Fisher) and 1.6 × 10^5^ OVCAR-3 cells were seeded in 24-well plates in OPTIMEM medium without FBS. After 24 h, medium with serum was added and phenotype was assessed 48 h post-transfection.

### 2.4. Western Blot Analysis

Cells were rinsed in ice-cold PBS, extracts were prepared in lysis buffer (50 mM Tris HCl pH 7.5, 1 mM EDTA, 1 mM EGTA, 1 mM NaV_3_O_4_, 10 mM sodium β-glycerophosphate, 50 mM NaF, 5 mM sodium pyrophosphate, 270 mM sucrose and 1% Triton X-100) supplemented with protease inhibitor cocktail (Roche) and 1 mM phenylmethylsulfonyl fluoride. Samples were electrophoresed in SDS-polyacrylamide gels and blotted onto nitrocellulose membranes. Blocked membranes were incubated overnight at 4 °C with the following antibodies: anti-human Hsp90 (1:1000; 610418, BD Biosciences, Barcelona, Spain); anti-SAMHD1 (1:2000; ab67820, Abcam, Cambridge, UK); anti-Cleaved PARP1 (E51) (1:1000, ab32064, Abcam); anti-GAPDH (1:10000, ab9485, Abcam); anti-pH2AX (Ser139) (1:1000, ab2577, Cell Signaling, Danvers, MA, USA) and anti-cleaved caspase 3 (Asp175) (1:1000, ab9661, Cell Signaling). After washing, the membranes were incubated with a secondary conjugated horseradish peroxidase antibody for 1 h at room temperature and then revealed with SuperSignal West Pico Chemiluminescent Substrate (Pierce Chemical, Rockford, IL, USA).

### 2.5. Cell Proliferation Assays

Cell proliferation capacity was measured by colony formation assays and growth curves. For colony formation assays, cells were trypsinized and plated in 6-well dishes at a density of 100 cells/well. Fifteen days later, cells were washed with PBS 1× and covered with fixing/staining solution (crystal violet solution + 20% ethanol + 2% formaldehyde 1×). Cells were washed several times with water and the number of colonies were counted. For growth curves, cells were plated in 96-well plates at 5000 cells/well and at indicated time points, cells were lysed with CellTiter-96^®^ AQueous One Solution Reagent (Promega, Madison, WI, USA) and luminescence was read using the Envision plate reader. All measurements were performed in triplicates.

### 2.6. Cell Cycle Analysis

Approximately 100,000 cells were harvested and washed with PBS. Cells were resuspended in 250 μL of Solution 10-Lysis buffer (ChemoMetec 910-3010) supplemented with 10 μg/mL of Solution 12–500 μg/mL DAPI (ChemoMetec 910-3012) and incubated at 37 °C for 5 min. Then, 250 μL of Solution 11-Stabilization buffer (ChemoMetec 910-3011) was added. Moreover, 10 μL of the cell suspension was load into the chambers of the NC-Slide A8. Cell cycle analysis was run in the NucleoCounter^®^ NC-3000™. 

### 2.7. Quantitative RT-Polymerase Chain Reaction (qRT-PCR)

For relative mRNA quantification, RNA was extracted using the NucleoSpin RNA II kit (Macherey-Nagel), as recommended by the manufacturer, including the DNase I treatment step. Reverse transcriptase was performed using the PrimeScript™ RT-PCR Kit (Takara, Shiga, Japan). Expression of genes related with molecular mechanisms of cancer was evaluated by using the commercial TaqMan™ Array Human Molecular Mechanisms of Cancer (4414161, Thermo Fisher), which included primers and probes for 96 genes. mRNA relative levels of all genes were measured by two-step quantitative RT-PCR and normalized to GAPDH mRNA expression using the DDCt method. Primers and DNA probes were purchased from Life Technologies TaqMan gene expression assays.

### 2.8. Drugs

The 1-beta-D-arabinofuranosylcytosine (AraC), difluorodeoxycytidine (gemcitabine), 5-chloropyrimidine-2,4(1H,3H)-dione (fluorouracil), and pemetrexed were purchased from Sigma-Aldrich (Madrid, Spain). Cisplatin and carboplatin were obtained from the hospital pharmacy service.

### 2.9. Evaluation of Cytotoxicity

Cells were treated at indicated doses of the test compounds for 4 days and the number of viable cells was quantified by a tetrazolium-based colorimetric method (MTT method) as described elsewhere 30. MTT assay measures metabolic activity of cells, resulting in a very sensitive procedure to evaluate cell viability and cell proliferation, including the effect of cytostatic agents that slow or stop cell growth.

### 2.10. Evaluation of DNA Damage and Apoptosis

Cells were treated with the corresponding drug concentrations or left untreated as a control for 24-h before evaluation of γ-H2AX by flow cytometry and western blot and PARP cleaved and cleaved caspase 3 by western blot, as described above. For flow cytometry staining, cells were permeabilized using Intracellular Staining Permeabilization Wash Buffer (BioLegend, San Diego, CA, USA) diluted in water prior to labeling with the Alexa Fluor^®^ 647 Mouse anti-H2AX (pS139) antibody for 30 min at 4 °C in dark. After incubation, cells were washed with Intracellular Staining Permeabilization Wash Buffer and fixed with 1% formaldehyde (FA) prior to acquisition in the flow cytometer. Flow cytometry was performed in a FACS LSRII flow cytometer (BD Biosciences). Data were analyzed using the FlowJo software (Single Cell Analysis Software, Ashland, OR, USA).

### 2.11. Statistical Analysis

A comprehensive cohort description analysis based on demographic, clinical, and biological data was performed. Categorical variables were summarized through frequencies and percentages, and quantitative variables using means and standard errors, or medians and interquartile ranges. Different endpoints were analyzed for each tumor: (i) Time to progression (TTP) was defined as the time from date of cohort treatment initiation to date of progression; (ii) overall survival (OS) was defined as the time from date of cohort treatment initiation to date of death resulting from any cause; (iii) disease-free survival (DFS) was defined as the time from curative therapy (surgery) to distance relapse; (iv) overall survival since cancer diagnosis (OSCD) was defined as the time from cancer diagnosis to date of death resulting from any cause. Patients who were alive (for OS) and disease-free (for TTP) will be censored at date of last follow-up. Median times for TTP, OS, DFS, and OSCD will be estimated using the method of Kaplan–Meier and reported with their confidence intervals (CI) at the 95% level. Log rank was used to compare Kaplan–Meier Curves. ORR will be reported as the proportion of patients who have a partial (PR) or complete response (CR) to therapy. 

Experimental data were analyzed with the PRISM statistical package If not stated otherwise, all data were normally distributed and expressed as mean ± SD of at least three independent experiments performed in duplicate. *p*-values were calculated using an unpaired, two-tailed, *t*-student test.

## 3. Results

### 3.1. SAMHD1 Is Differentially Expressed in Solid Tumors and Correlates with Tumor Differentiation or Grade

To determine the contribution of SAMHD1 expression to cancer progression and/or treatment efficacy, available primary or metastatic tumor biopsies were retrospectively collected for all patients included in the study. Expression of SAMHD1 across different tumor types was evaluated by immunohistochemistry (Figure 1) and samples were stratified according to SAMHD1 expression. SAMHD1 expression varied significantly across tumor types, ranging from high percentage of positivity in rectal to low expression in pancreatic tumors, whereas values of 50–60% positivity were obtained for ovarian, NSCLC and breast cancer cases (Table 2). Our data do not differ from that available in public databases (Appendix A), except for rectal cancer, where in our series SAMHD1 was strongly expressed in all neoplastic cells. Considering the lack of variability in SAMHD1 expression levels among rectal and pancreatic cancer cases, these two cancer types were excluded from further analysis, as the absence of a control arm made it impossible to evaluate SAMHD1 function in these tumor types.

As SAMHD1 function might influence cell proliferation, we first investigated whether SAMHD1 expression was associated with tumor differentiation grade or histologic type as a surrogate measure of tumor cell proliferation. SAMHD1 positivity correlated with poorly differentiated histology (*p* = 0.024) and high grade (*p* = 0.011) in NSCLC. In ovarian carcinoma samples, those with high grade serous papillary ovarian carcinoma were the most positive for SAMHD1 (*p* = 0.028). For breast cancer patients, SAMHD1 positivity was correlated with high grade breast tumors (*p* = 0.017) (Table 3).

### 3.2. SAMHD1 Expression as a Negative Prognostic Factor in Breast, Ovarian, and NSCLC Patients

The contribution of SAMHD1 in cancer onset and progression is controversial, with strong evidence demonstrating both its role as a tumor suppressor and as an oncogene [34]. Thus, to gain insight into the putative role of SAMHD1 as a prognostic factor in solid tumors, we first evaluated disease-free survival (DFS) in ovarian, NSCLC, and breast cohorts. SAMHD1 positive patients presented shorter DFS than SAMHD1 negative patients in all three cohorts of patients (Figure 2A). 

Ovarian carcinoma patients showed a median DFS of 52 months for SAMHD1 negative patients in front of 11 months for SAMHD1 positive patients (*p* = 0.005). Median DFS in NSCLC cancer patients was 22 months for SAMHD1 negative patients compared to 5 months for SAMHD1 positive patients (*p* = 0.009). In breast cancer patients, median DFS was 64 months for SAMHD1 negative patients compared to 21 months for SAMHD1 positive patients (*p* = 0.001). Moreover, when we exclude patients diagnosed with advanced disease that did not receive any type of local therapy, SAMHD1 positivity continued to be associated with shorter DFS in all the cohorts (Appendix A). 

The multivariate analysis for breast and NSCLC patients showed that negative SAMHD1 status was the only factor significantly associated with longer DFS (*p* = 0.005 and *p* = 0.04, respectively) and a similar trend was observed in ovarian cancer patients (longer DFS associated to SAMHD1 negativity (*p* = 0.09)) (Appendix A).

Accordingly, when we evaluated overall survival since cancer diagnosis (OSCD) patients with SAMHD1 positive tumors presented shorter OSCD than negative patients in all three cohorts (Figure 2B). Median OSCD was 157 months in SAMHD1 negative in front of 62 months in SAMHD1 positive, for ovarian cancer patients (log rank function, *p* = 0.040). NSCLC patients with SAMHD1 negative presented a median OSCD of 36 months in front of 14 months in SAMHD1 positive patients, (log rank function, *p* = 0.004). Finally, the median OSCD for SAMHD1 negative breast cancer patients was 116.7 months in front of 65.9 months for SAMHD1 positive (log rank function, *p* = 0.004). 

Overall, these data indicate that SAMHD1 expression is a strong independent negative prognostic factor in ovarian, breast and NSCLC patients.

### 3.3. Predictive Significance of SAMHD1 Expression in Cancer Patients Treated with Antimetabolite- and/or Platin-Containing Regimens

SAMHD1 function has been clearly associated with the efficacy of several nucleoside analogs used as antivirals or chemotherapeutic agents. Thus, we determined the value of SAMHD1 as a predictive factor in ovarian, NSCLC and breast cancer treated with corresponding antimetabolite-containing regimens. As above, tumor biopsies were stratified as positive or negative and overall response rate (ORR) was evaluated for each tumor. Interestingly, ovarian and NSCLC cancer patients treated with antimetabolite plus platinum-based chemotherapeutic regimens presented lower ORR in case of SAMHD1 positivity (*p* = 0.04 and *p* = 0.016, respectively, Table 4). On the contrary, no association was found between SAMHD1 expression and treatment efficacy in breast cancer patients treated with capecitabine (*p* = 0.232, Table 4).

In line with ORR data, when time to progression since therapy initiation was evaluated, SAMHD1 positive patients presented shorter TTP than SAMHD1 negative patients, for ovarian and NSCLC tumors (log rank function, *p* = 0.003 and *p* = 0.005, respectively). In contrast, no differences were observed in TTP related to SAMHD1 status for breast cancer patients (log rank function, *p* = 0.511) (Figure 3A). Similar results were obtained with OS since therapy initiation was evaluated; a shorter OS was observed in SAMHD1 positive patients than in SAMHD1 negative patients, only for ovarian and NSCLC tumors (log rank function, *p* = 0.060 and *p* = 0.014, respectively). Still, no differences in OS related to SAMHD1 status for breast cancer patients were observed (log rank function, *p* = 0.676) (Figure 3B). These data suggest that SAMHD1 may serve as a predictive factor only in NSCLC and ovarian cancer but not in breast cancer. Interestingly, NSCLC and ovarian patients have been treated with antimetabolites in combination with platinum-containing agents in contrast with breast patients that had been treated with capecitabine alone.

### 3.4. Loss of SAMHD1 Induced Cellular Apoptosis by Enhanced Genomic Instability and DNA Damage Response

To unravel the molecular mechanisms associated to the prognostic and predictive value of SAMHD1 expression in the clinic, in vitro models of SAMHD1 depletion were developed in breast and ovarian cell lines. Initial evaluation of SAMHD1 KO breast cancer cells did not show any difference in cell proliferation capacity compared to wild type cells, neither when cell growth was evaluated (Appendix A) nor in colony formation assays. Similarly, no relevant differences were observed between wild type and KO cells in cell cycle profile and expression of main genes and pathways associated to molecular mechanisms of cancer, except for the different expression of SAMHD1 protein (Appendix A). 

As SAMHD1 has been shown to significantly affect antiviral and cytotoxic efficacy of several antimetabolites, cell proliferation capacity was also measured in SAMHD1 KO cells treated with the chemotherapeutic drugs used in the clinical cohorts, i.e., gemcitabine, pemetrexed and 5-fluorouracil, the pharmacologically active drug of capecitabine and the platinum-based drugs, cisplatin, and carboplatin. Ara-C was added as a positive control, as it has been previously shown that SAMHD1 significantly impairs Ara-C efficacy in vitro and in vivo, by directly hydrolyzing the triphosphorylated form of the drug [3]. Again, no differences were observed in the proliferation capacity of SAMHD1-KO cells in the presence of drugs except for Ara-C, used as a control (Appendix A), suggesting that SAMHD1 expression is not directly affecting cell proliferation capacity irrespective of the treatment.

Independently of SAMHD1 canonical dNTP hydrolase activity, numerous evidences indicate that SAMHD1 is also involved in the DNA damage response (DDR) [35], a process that may also contribute to cancer onset, disease progression and certain therapies affecting cell proliferation and survival following DNA damage induction [29]. Thus, we evaluated DNA damage induction and survival in wild type and KO in vitro model by assessing γH2AX expression—a well-known marker for DNA double stranded breaks (DSBs)—by flow cytometry, which has been widely used as a sensitive and reliable method for quantification of the DNA damage response [36,37,38]. SAMHD1 KO cells tend to show higher levels of γH2AX than wild type cells (Figure 4A), both when measured and quantified by flow cytometry and also confirmed by western blot (Appendix A). To test whether this increased in DNA damage in KO cells is translated into increased apoptosis levels, we evaluated cleaved caspase 3 and cleaved PARP protein expression by western blot. Our results showed increased amounts of PARP cleaved and cleaved caspase 3 in SAMHD1 KO cells, suggesting increased apoptosis in SAMHD1-depleted cells (Figure 4B and Appendix A). 

Then, DNA damage induction and apoptosis were evaluated after treatment with the DNA damage inducers cisplatin and carboplatin, or the antimetabolite fluorouracil, used in the study cohorts in combination with antimetabolites. Interestingly, SAMHD1 KO cells showed a trend to increased expression of both DNA damage and apoptotic markers compared to wild type cells when treated with platinum derivatives (Figure 4C,D and Appendix A). In contrast, no effect was seen when cells were treated with fluorouracil, a nucleotide analogue that blocks thymidylate synthase impeding DNA replication. Similar results were obtained in ovarian cancer cells after effective know-down of SAMHD1 by RNA interference (Appendix AA), showing increased DNA damage and apoptosis upon SAMHD1 depletion an effect that was further enhanced upon treatment with platinum derivatives (Appendix A). Overall, in vitro data support the idea that SAMHD1 depletion is enhancing susceptibility to DNA damage and, therefore, providing biological basis for the observed SAMHD1 predictive value in patient cohorts.

## 4. Discussion

Cancer is a major burden of disease worldwide. According to World Health Organization data, in 2018, there were an estimated 18.1 million new cases and 9.6 million deaths from cancer [39]. Although cancer incidence and mortality vary according to the tumor type, advanced-stage tumors are mostly incurable. To determine the best action for cancer patient care, an early diagnosis is essential. Moreover, it is even more critical to identify biomarkers that can predict disease evolution soon after diagnosis. In this regard, new biological markers allow, on the one hand, to better assess the prognosis of patients in early stages, and on the other hand, to predict the response to available treatments and the development of new drugs. 

In recent years, growing evidence pointed towards SAMHD1 as one of these putatively valuable biomarkers. Its role is a matter of intense debate, mainly in different hematological cancers and to a lesser extent in solid tumors such as NSCLC and colorectal cancer [34]. Here, we performed the first in-depth study of SAMHD1 role in advanced disease cases of solid tumors. There is still the need to develop effective biomarkers and more efficient alternative treatment options. Our study included 98 advanced cancer cases of high incident and/or high mortality tumors such as NSCLC, breast, ovarian, and pancreatic cancers, on which SAMHD1′s role has been extensively evaluated. Interestingly, we found significant heterogeneity in SAMHD1 expression across different tumor types, from 100% positivity for rectal cancer, through 68% in ovary, 61% in breast, 50% in NSCLC and 11% in pancreatic cancer, similar to available data from public databases, except for rectal cancer, where the distinct methodology used for its evaluation may influence in the high positivity observed. In line with tumor data, significant heterogeneity in SAMHD1 expression has also been reported in normal human tissues, being ubiquitously expressed in lymphoid cells [40]. Of note, in all analyzed tissue biopsies, infiltrating cells from immune origin present the highest expression. Here, we found that SAMHD1 expression was significantly associated with tumor histology and tumor grade, being poorly differentiated high-grade tumors those presenting the highest proportion of SAMHD1 positive cases. These data point towards a relevant role for SAMHD1 in cell proliferation and cancer development in solid tumors as previously suggested [34], albeit somewhat contradictory to existing in vitro and ex vivo data where it was assumed that SAMHD1 depletion may favor cell proliferation, consequence of the dysregulation of intracellular dNTP pool [41,42]. However, although the reduction of dNTP levels increases the fidelity of DNA polymerases, it can also lead to stalling of replication forks, the accumulation of single-stranded DNA, and chromosomal rearrangements. Consistent with these outcomes, decreased dNTP pools have been proposed to be a source of genomic instability in early stages of cancer development [43]. Indeed, constitutively high dNTP levels affect cell cycle progression delaying entry into S phase [44,45]. In this process, SAMHD1 function has been linked by promoting S phase progression through the regulation of cellular dNTP availability [46]. SAMHD1 downregulation has also been linked to protection of cancer development in patients [6,47,48] casting doubt on the role of SAMHD1 as a tumor suppressor and highlighting the need of further evaluating its clinical significance in different tumor types. 

Here, we found that SAMHD1 expression was significantly associated with poorer prognostic clinical outcomes, including DFS and OSCD in all tumor types analyzed, including advanced disease cases of breast, ovarian, and NSCLC cancer. Interestingly, SAMHD1 positivity was an independent prognostic factor of worst DFS in breast and NSCLC cancer. Collectively, our data confirm the key role of SAMHD1 in cancer progression but also indicates that its function might depend on the specific tumor type, being able to act as a tumor suppressor, as reported previously mainly for hematological tumors [6,7,49] or alternatively as a promoter of cancer progression. Indeed, data derived from TCGA or ICGC databases show similar results, pointing towards SAMHD1 tumor promoter function in ovarian carcinoma, but not in lung and breast carcinoma [48,50,51], in contrast with the data presented here. Differences in methods for determining SAMHD1 status, the heterogeneity of the clinical series evaluated, and the possible different treatments that patients receive may explain the apparent contradictory results between data from databases and our data, which rely on rather small but homogenous cohorts evaluated using the same criteria across tumor types and samples. Overall, our results, and that of others, point toward a tumor type-dependent function of SAMHD1 in cancer onset and/or progression, reflecting once more the great heterogeneity of cancer biology, which deeply challenges the drive for personalized treatment. 

The clinical impact of SAMHD1 in advanced solid tumors has been evaluated in a limited number of studies. In this sense, our data are in accordance with previous results in (i) colorectal cancer where high SAMHD1 expression level in tumors correlated to increased risk of metastases [15], (ii) in untreated classical Hodgkin lymphoma, where SAMHD1 was an independent adverse prognostic factor [52] and also in (iii) NSCLC EGFR mutated cancer patients where SAMHD1 serum levels were significantly increased when compared with normal control, upon cancer progression [16]. 

Previous studies have demonstrated that SAMHD1 regulates the therapeutic efficacy of nucleotide analogs used as antivirals or as antineoplastic agents both in vitro and in vivo in AML. Thus, we evaluated the predictive role of SAMHD1 in the response to antimetabolite-containing treatment regimens, evaluating clinical response and TTP. For BC patients treated with capecitabine, we did not find any correlation between SAMHD1 status and ORR or TTP. On the contrary, when we analyzed NSCLC patients treated with platinum in combination with gemcitabine or pemetrexed, or with gemcitabine in monotherapy, and ovarian cancer patients treated with cisplatin or carboplatin in combination with gemcitabine or gemcitabine in monotherapy. SAMHD1 positivity was associated with lower ORR and lower TTP, following the same trend observed for prognostic value and suggesting a role of SAMHD1 as a predictor of poorer outcome in these subsets of patients. 

Similar to our data, SAMHD1 effect on antineoplastic drug efficacy was first demonstrated in vivo with cytarabine, the standard treatment for AML, where SAMHD1 expression results in limited cytarabine efficacy due to enhanced hydrolysis of the active metabolite [19,20]. Additional evidence pointed towards a similar effect for other antiproliferative drugs used to treat hematological cancers as fludarabine, decitabine, vidarabine, and clofarabine, all considered substrates of SAMHD1 [53], or forodesine an inhibitor of dGTP synthesis [54]. However, none of the treatments of the cohorts here evaluated can be directly hydrolyzed by SAMHD1, as either they are not nucleotide analogues as pemetrexed. They need additional steps of enzymatic processing as is the case of capecitabine, the prodrug of 5-FU, or have been reported as functional inhibitors of SAMHD1 as gemcitabine [55], indicating that predictive value of SAMHD1 is not limited to drugs that can be used as enzyme substrates but also goes far beyond its effects in hematological cancers. It has to be considered that, in a significant proportion of patients, standard treatment includes the combination of antimetabolites with other chemotherapeutic drugs, introducing additional layers of complexity that may also affect SAMHD1 predictive value. Further studies in larger cohorts of specific tumors are needed to better determine the specific contribution of each treatment schedule. 

Overall, our data provide strong evidence of the involvement of SAMHD1 function in advanced cancer. However, the underlying mechanisms of SAMHD1 in the induction and regulation of tumorigenesis remain unknown, although it is hypothesized that SAMHD1 may mediate cell proliferation via both dNTPase-dependent or -independent mechanisms. Our data do not support the idea that SAMHD1 depletion provides transformed cells with a growth advantage simply due to elevated dNTP levels. Alternatively, the role of SAMHD1 in cancer may relate to its functions in DNA repair and DNA replication, which are independent of dNTP degradation [6,35,56]. Consistent with this hypothesis, we observed that SAMHD1 KO cells present enhanced susceptibility to DNA damage leading to increased apoptosis, an effect that is significantly enhanced upon in vitro treatment with DNA damaging agents. Indeed, SAMHD1 is known to play a direct role in genome maintenance by promoting DNA end resection to facilitate DNA DSB repair by homologous recombination [35], and also participate in the degradation of nascent DNA at stalk replication forks in human cell lines, allowing the forks to restart replication and promoting cell survival [56]. Both shreds of evidence support our results, demonstrating that SAMHD1 plays a critical role in the response of cancer cells to DSB-inducing agents as platinum derivatives. 

## 5. Conclusions

In summary, our data provide strong evidence of SAMHD1′s prognostic and predictive value in advanced cancers, which may be driven by SAMHD1′s role in DNA damage function. Based on both the clinical importance of SAMHD1 and the detailed knowledge of its functions and regulation mechanisms, SAMHD1 has become an attractive target for cancer treatment. To date, direct inhibitors of SAMHD1 dNTPase activity have been proposed [3,54], and combined therapies leading to the inhibition or enhancement of SAMHD1 activity by regulating phosphorylation through CDK inhibitors or acetylation have been successfully applied in in vitro models [22,57,58]. One of the potential opportunities in oncology is to find targets that small molecules can target to achieve an effect on the tumor, as this may be the case of SAMHD1. Further evaluation in additional cohorts, retrospective and prospective clinical studies, including combination therapies that modulate SAMHD1 expression and function, will provide definite proof of its value and potential clinical applications. 

## Figures and Tables

**Figure 1 cancers-14-00641-f001:**
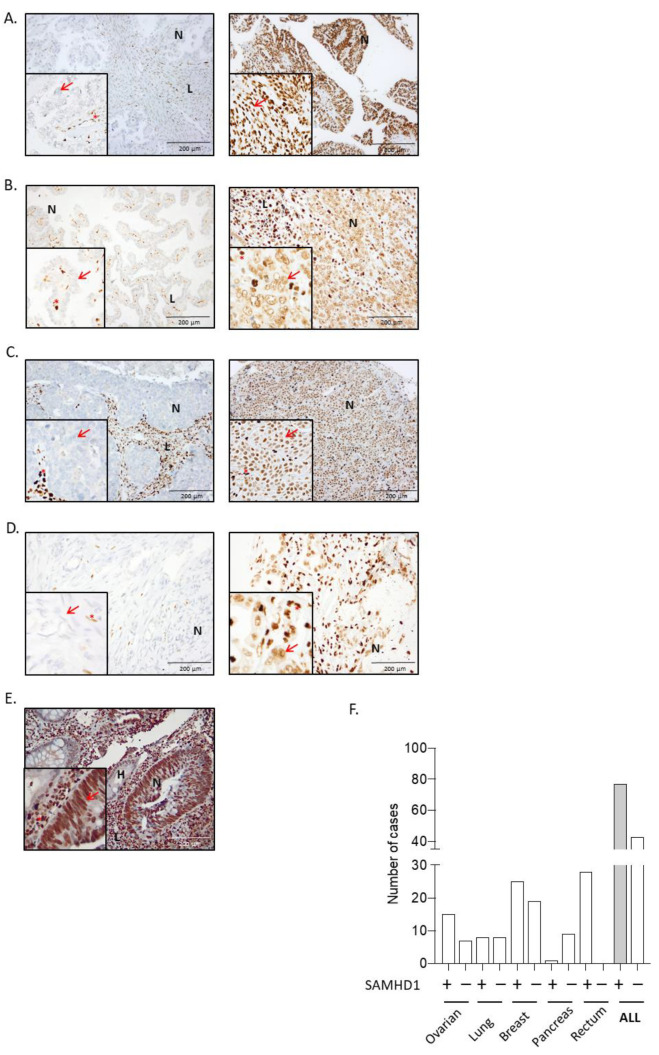
Expression of SAMHD1 protein by immunochemistry in tumor samples. (**A**–**E**) Representative microscopy images of SAMHD1 expression in paraffin-embedded tumor biopsies for the different tumor types included in the study, from ovarian (**A**), NSCLC (**B**), breast (**C**), pancreas (**D**), and rectal (**E**). Images on the left represent negative SAMHD1 expressing tumors and on the right positive expressing tumors in all cases except in rectum (**E**), where all analyzed tumors presented extremely high levels of SAMHD1 expression. High expression of SAMHD1 observed in lymphocytic cells infiltrating in the tumors was used as a positive control of immunohistochemistry for negative or low expressing biopsies. In case of negative or low expressing biopsies, high expression of SAMHD1 was observed in lymphocytic cells. (**F**) Summary of all analyzed biopsies stratified based on SAMHD1 expression in positive or negative (cutoff >25% of tumor cells). N, neoplastic cells; L, lymphocytes, H, healthy tissue. In higher magnification images, red arrows indicate neoplastic cells and red asterisks lymphocytic cells. SAMHD1 expression was exclusively nuclear.

**Figure 2 cancers-14-00641-f002:**
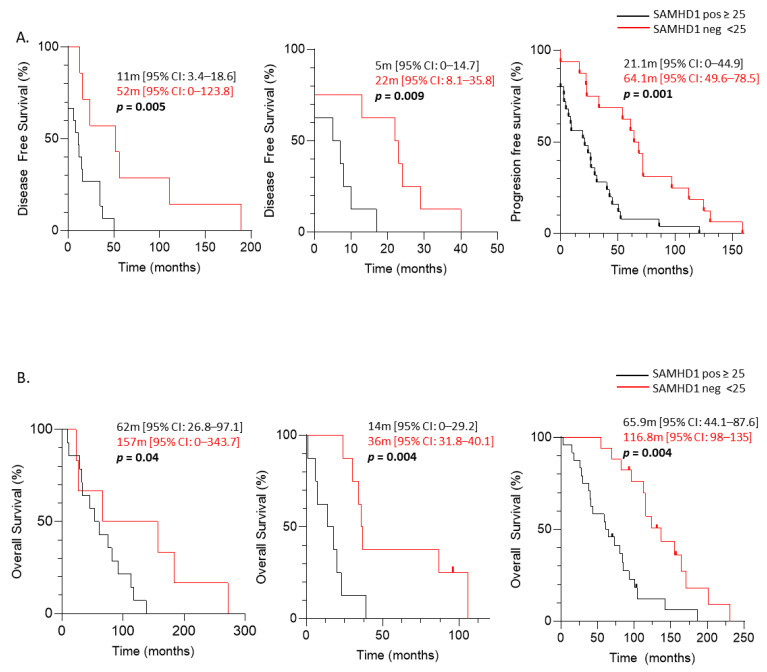
Prognostic value of SAMHD1 in ovarian, lung and breast cancer cohorts. Kaplan–Meier curves of disease-free survival (DFS) (**A**) and OS since cancer diagnostic (**B**) according to SAMHD1 status for different tumor types. From left to right: ovarian, NSCLC and breast. Kaplan–Meier curves are represented. SAMHD1 expression below 25% in cancer cells was considered as negative SAMHD1 (red lines) and equal or above 25% was considered as positive SAMHD1 tumors (black lines). Median survival times with CI 95% of both groups are showed. Log rank test was used to test the significance and censored patients are indicated by vertical line.

**Figure 3 cancers-14-00641-f003:**
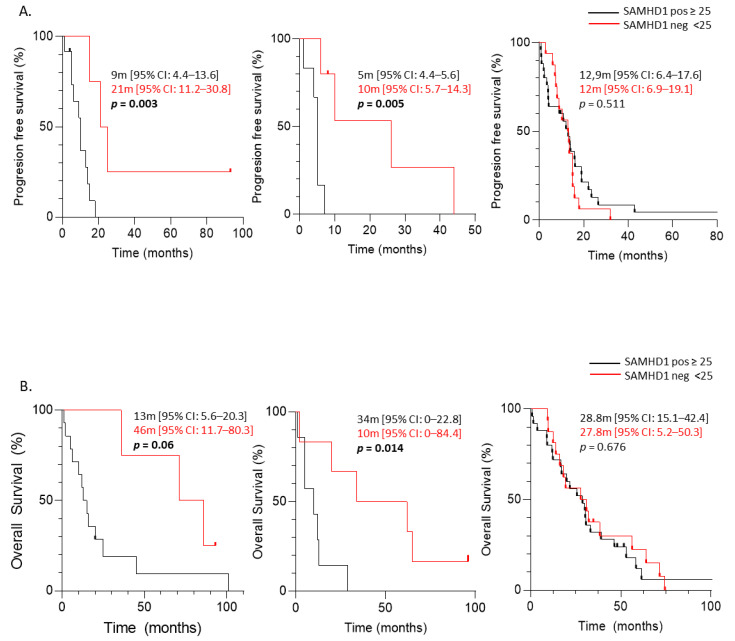
Predictive value of SAMHD1 in ovarian, lung and breast cancer cohorts treated with antimetabolite-containing regimens. Kaplan–Meier curves of time to progression (TTP) (**A**) and overall survival since cohort treatment initiation (OS) (**B**) for each tumor according to SAMHD1 status for different tumor types. From left to right: ovarian, NSCLC and breast. SAMHD1 expression below 25% in cancer cells was considered as negative SAMHD1 (red lines) and equal or above 25% was considered as positive SAMHD1 tumors (black lines). Median survival times with CI 95% of both groups are showed. Log rank test was used to test the significance and censored patients are indicated by vertical line.

**Figure 4 cancers-14-00641-f004:**
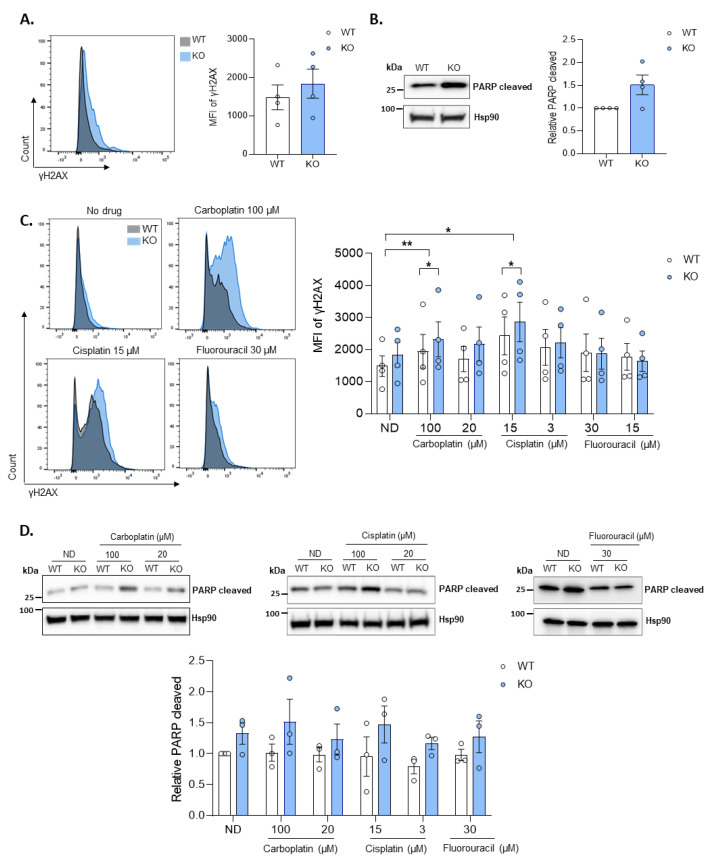
SAMHD1-depletion induces DNA damage and apoptosis after treatment with platinum derivatives. (**A**) The DNA damage marker γH2AX expression in WT and SAMHD1-KO cell lines. Representative flow cytometry histogram (left) with overlay of WT (grey) and SAMHD1-KO (blue) T47D cells comparing γH2AX expression. Histogram has been normalized to the modal values. Bar graph (right) showing mean fluorescence intensity (MFI) of γH2AX in WT and SAMHD1-KO T47D cells. Mean ± SEM of four independent experiments is shown. (**B**) Cell apoptosis measured as PARP cleaved and cleaved caspase 3 in WT and SAMHD1-KO cell lines. Representative western blot (left) and quantification (right) showing differential expression of PARP cleaved and cleaved caspase 3 in T47D WT and SAMHD1-KO cells. Mean ± SEM of four independent experiments is shown. (**C**) Left panel, representative flow cytometry histograms measuring γH2AX expression in WT (grey) and SAMHD1-KO (blue) T47D cells treated with carboplatin, cisplatin, and fluorouracil for 24 h. Right panel, bar graphs representing MFI of γH2AX expression in WT and KO T47D cells after treatment with different concentrations of carboplatin, cisplatin, and fluorouracil. Mean ± SEM of four different experiments is shown. (**D**) Representative western blots (upper panel) and quantification (bottom panel) showing PARP cleaved and cleaved caspase 3 expression in WT and SAMHD1-KO T47D cells treated with carboplatin, cisplatin and fluorouracil for 24 h. Mean ± SEM of four different experiments is shown. *, *p* < 0.05; **, *p* < 0.01.

**Table 1 cancers-14-00641-t001:** Clinical characteristics of patients and tumors.

Variable	Type of Tumor				
	Rectal (*n* = 30)	Ovarian (*n* = 22)	Lung (*n* = 16)	Breast (*n* = 46)	Pancreas(*n* = 14)
**Age** (y), Mean Interquartile range (IQR)	62.77 (37–80)	62.82 (51–82)	61.88 (47–83)	54.02 (29–84)	63.57 (45–80)
**Gender**					
Male, *n* (%) Female, *n* (%)	19(63.3%) 11(36.7%)	- 22 (100%)	13 (81.3%) 3 (18.7%)	- 46 (100%)	9 (64.3%) 5 (35.7%)
**Line of Therapy**					
Neoadjuvant, *n* (%)	30(100%)				
First, *n* (%)		1(4.5%)		15 (32.6%)	13 (92.9%)
Second, *n* (%)		7 (31.8%)	16 (100%)	15 (32.6%)	1 (7.1%)
Third, *n* (%)		6 (27.3%)		8 (17.4%)	
≥Fourth, *n* (%)		8 (36.4%)		8 (17.4%)	
**Overall Response Rate (ORR)** *					
Complete response (CR), *n* (%)		3(13.6%)	0 (0%)	3 (18.7%)	
Partial response (PR), *n* (%)	6 (20%)	8 (36.4%)	5 (31.3%)	1 (2.2%)	
Stable disease (SD), *n* (%)	21(70%)	3 (13.6%)	6 (37.5%)	17 (37%)	2 (14.3%)
Progressive disease (PD), *n* (%)	2 (6.7%)	5 (22.8%)	2 (12.5%)	15 (32.6%)	4 (28.6%)
Non-evaluable (NE), *n* (%)	1 (3.3%)	3 (13.6%)		13 (28.2%)	8 (57.1%)
TTP #, months, median (IQR)	26.00 (0.00–56.79)	10.00 (4.68–15.31)	6.00 (2.75–9.23)	12.97 (9.06–16.89)	6 (4.3–7.46)
OS £, months, median (IQR)	77.85 (68.11–87.59)	18.00 (9.68–26.31)	13.00 (1.25–24.74)	28.81 (18.95–38.67)	11.00 (7.33–14.66)
**DFS &**, months, median (IQR)					
All	nd	15.00 (10.40–19.59)	8 (2.1–13.88)	29.83 (17.83–41.82)	6 (4.3–7.46)
Excluding “de novo” ¥	nd	24.00 (7.86–40.13)	17 (4.05–29.94)	40.41 (22.77–58.04)	15 (11.90–18.09)
OSCD α, months, median (IQR)	nd	66 (33.03–98.96)	24.00 10.28–37.72	93.83 72.33–115.32	17.00 0–51.83

IQR, interquartile range; * ORR: the proportion of patients who have a partial (PR) or complete response (CR) to therapy; # TTP: time to treatment progression, the time from date of treatment initiation to date of progression or death resulting from any cause (whichever occurred first); £ OS: the time from date of cohort treatment initiation to date of death resulting from any cause; & DFS: disease-free interval, the time from curative therapy (surgery) to distance relapse; ¥ Excluding “de novo”: excluding patients with initial advanced disease without further resection (DFS = 0); α OSCD: overall survival since cancer diagnosis, the time from cancer diagnosis to date of death resulting from any cause; nd. not determined.

**Table 2 cancers-14-00641-t002:** Expression of SAMHD1 by immunohistochemistry in patient biopsies across tumor types.

SAMHD1 Expression	Type of Tumor (*n*)				
	Rectal (*n* = 30)	Ovarian (*n* = 22)	Lung (*n* = 16)	Breast (*n* = 46)	Pancreas (*n* = 14)
**% Expression,** mean (IQR)	64.64 (25–90)	55.82 (0–100)	34.06 (0–100)	27.88 (0–80)	16.11 (0–75)
**Positivity ratio evaluable patients,***n* (%)	28 (100%)	15 (68.2%)	8 (50%)	25 (61%)	1 (11.1%)
**Non evaluable,** *n*	2	0	0	5	5

**Table 3 cancers-14-00641-t003:** SAMHD1 expression and tumor histology and grade of ovarian, lung and breast cancer patients.

Tumor	Variable	SAMHD1 Positivity Ratio		*p*-Value *
		Positive (≥25%)	Negative (<25%)	
**Ovarian**	**Histology** High grade serous papillary, *n* (%) Clear cell carcinoma, *n* (%) Low grade serous papillary, *n* (%)	14 (82.3%) 1 (25%) 0 (0%)	3 (17.7%) 3 (75%) 1 (100%)	0.028 *
**Lung**	**Histology** Squamous, *n* (%) Adenocarcinoma, *n* (%) Poorly differentiated, *n* (%)	2 (33.4%) 1 (20%) 5 (100%)	4 (66.6%) 4 (80%) 0 (0%)	0.024 *
	**Tumor grade** I, *n* (%) II, *n* (%) III, *n* (%)	1 (100%) 1 (14.3%) 7 (87.5%)	0 (0%) 6 (85.7%) 1 (12.5%)	0.011 *
**Breast**	**Tumor grade**^£^ I, *n* (%) II, *n* (%) III, *n* (%)	0 (%) 8 (40%) 16 (84.2%)	0 (0%) 12 (60%) 3 (15. 8%)	0.017 *

* Pearson’s chi-squared; ^£^ tumor grade was not evaluable in 2 of 41 patients.

**Table 4 cancers-14-00641-t004:** Response to treatment in ovarian, lung, and breast cancer patients depending on SAMHD1 positivity in the corresponding tumor biopsies.

Variable	Type of Tumor (*n*)					
	Ovarian (22)		Lung (16)		Breast (46)	
	Pos. (≥25)	Neg. (<25)	Pos. (≥25)	Neg. (<25)	Pos. (≥25)	Neg. (<25)
**Clinical response rate**						
Complete response (CR), *n* (%)	2 (66.6%)	1 (33.4%)	0 (0%)	0 (0%)	1 (100%)	0 (0%)
Partial response (PR), *n* (%)	4 (50%)	4 (50%)	0 (0%)	5 (100%)	11 (68.8%)	5 (31.2%)
Stable disease (SD), *n* (%)	3 (100%)	0 (0%)	4 (60%)	2 (40%)	2 (16.8%)	10 (83.2%)
Progressive disease (PD), *n* (%)	5 (100%)	0 (0%)	2 (100%)	0 (0%)	11 (91.6%)	1 (8.4%)
Overall response rate (ORR) √						
Yes, *n* (%)	6 (54.5.5%)	5 (45.5%)	0 (0%)	5 (100%)	12 (70.5%)	5 (29.5%)
No, *n* (%)	8 (100%)	0 (0%)	6 (75%)	2 (25%)	13 (54.2%)	11 (45.8%)
*p*-value *	0.04 *		0.016 *		0.232	

* Pearson’s chi-squared; √ overall response rate: CR+PR; Pos.: positive; Neg.: negative.

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
