# Peer review of "Modulation of DNA Damage Response by SAM and HD Domain Containing Deoxynucleoside Triphosphate Triphosphohydrolase (SAMHD1) Determines Prognosis and Treatment Efficacy in Different Solid Tumor Types"

_cancers, 2022, doi:10.3390/cancers14030641_

Round 1

Reviewer 1 Report

The article is now ready for publication. No further changes are necessary.

Author Response

The article is now ready for publication. No further changes are necessary.

Thanks for the revision and comments.

Reviewer 2 Report

The authors made an effort to address most of the issues. However, supplemental Fig. 7D should be removed and results should be amended: the presented results are not convincing as there is no difference between siNT and siSAMHD1 and no differences upon treatment.

Author Response

Reviewer 2

The authors made an effort to address most of the issues. However, supplemental Fig. 7D should be removed and results should be amended: the presented results are not convincing as there is no difference between siNT and siSAMHD1 and no differences upon treatment.

Following the reviewer’s suggestion we have removed supplemental figure 7D and changed the results accordingly.

This manuscript is a resubmission of an earlier submission. The following is a list of the peer review reports and author responses from that submission.

Round 1

Reviewer 1 Report

The name of the gene must be provided in full before using the abbreviation the first time it appears: SAM and HD domain containing deoxynucleoside triphosphate triphosphohydrolase (SAMHD1). Source: NIH/Gene ID 25939.

The abstract

In a paper like this, the abstract should be a structured one containing the following subtitles: Background / Objectives / Material and Methods / Results / Conclusions.

In the introduction of the paper, the authors do not describe what the SAMHD1 protein does in normal cells. They should explain the functions of this protein and how it keeps the pool of dNTs at the expected level. Furthermore, they do not explain the biochemical reaction SAMHD1 catalyzes.

See

Ji, X., Tang, C., Zhao, Q., Wang, W., & Xiong, Y. (2014). Structural basis of cellular dNTP regulation by SAMHD1. Proceedings of the National Academy of Sciences111(41), E4305-E4314.

Arnold, L. H., Groom, H. C., Kunzelmann, S., Schwefel, D., Caswell, S. J., Ordonez, P., ... & Bishop, K. N. (2015). Phospho-dependent regulation of SAMHD1 oligomerisation couples catalysis and restriction. PLoS pathogens11(10), e1005194.

Line 194: authors say High expression of SAMHD1 observed in lymphocytic cells infiltrating in the tumors was used as a positive control of immunohistochemistry for negative or low expressing biopsies.

This sentence is not clear. They used the infiltrating lymphocyte positivity as a control of the same tumor?

Please give the full name of IQR ,(interquartile range), when it is first used.

Table 2 is not clear enough. What do the authors mean with relative expression?

The following paragraph should be rephrased making it more reader friendly.

SAMHD1 positivity correlated with poorly differentiated histology (p=.024) and high differentiation grade (p=.011)  in NSCLC, with high grade serous papillary ovarian carcinoma (p=.028), and with poorly differentiation grade in breast tumors (p=.017), suggesting that SAMHD1 was indeed affecting cell proliferation capacity, albeit interestingly, SAMHD1 positivity was associated to less differentiated more aggressive tumors (

I really do not understand how SAMHD1 positivity correlated at the same time with poorly differentiated histology and high differentiation grade in NSCLC. Probably there is some error here. Or the word differentiation is incorrectly used. May be it should be

SAMHD1 positivity correlated with

NSCLC:  poorly differentiated histology (p=.024) and high  grade (p=.011)

Ovarian serous papillary carcinoma : high grade (p=.028),

Breast cancer: poor differentiation(p=.017),

suggesting that SAMHD1 was indeed affecting cell proliferation capacity, albeit interestingly, SAMHD1 positivity was associated to less differentiated more aggressive tumors (

The conclusion at the end of the paragraph is not coherent with what is said in the paragraph.

Line 314. It would be convenient to replace disease free interval for progression free survival

.

Lines 310 to 324 should be replaced by a Table. As it is now, it is not easy for the reader.

The following sentence must be rephrased fully /Line 339

Accordingly, when we evaluated overall survival since cancer diagnosis (OSCD) patients with positivity for SAMHD1 presented shorter OSCD than negative patients in all 340 three cohorts (Figure 2B), presenting a median OSCD of 157 months in SAMHD1 negative (95%CI, 0 to 343.6) in front of 62 months in SAMHD1 positive (95%CI, 26.8 to 97.1), for  ovarian cancer patients, (log Rank function, p=.040); also for NSCLC patients with a median OSCD of 36 months (95%CI, 31.8 to 40.1) in front of 14 months (95%CI, 0 to 29.2), respectively (log Rank function, p=.004); and for breast cancer patients with a median  OSCD of 116.7 months (95%CI, 98.2 to 135.2) in front of 65.9 months (95%CI, 44.1 to 87.6), 346 respectively (log Rank function, p=.004).

122 words in one sentence makes it quite difficult to be understood by any average reader, including this reviewer.

Line 369 says Similar results were obtained when OS  since therapy initiation was evaluated; a shorter OS

It should say Similar results were obtained with OS since therapy initiation was evaluated.

In Line 488 the authors say Collectively, our data confirm the key role of SAMHD1 in cell transformation and tumorigenesis but also…

This paper does not confirm the role of SAMHD1 neither in cell transformation nor tumorigenesis. In any case this paper hints to a role in progression. This is a different concept than transformation or tumorigenesis. They do not prove that SAMHD1 has anything to do with tumor initiation.

There are many papers that sustain exactly the opposite . The following paragraph was extracted from Coggins, S. A. A., Mahboubi, B., Schinazi, R. F., & Kim, B. (2020). SAMHD1 functions and human diseases. Viruses12(4), 382.

 Conversely, the literature shows that overexpression of SAMHD1 is associated with reduced cell proliferation likely due to the depletion of dNTPs necessary to properly replicate genomic DNA.

Line 285 It is not clear why rectal and pancreatic cancers are excluded from consideration. The fact that they do not change survival or progression free survival because according to the authors they are always overexpressed is a poor criteria for exclusion.

The 25% positivity cut off for SAMHD1 seems excessively low.

Reviewer 2 Report

The authors provide insights into the prognostic and predictive value of SAMHD1 expression in different solid tumors. They analyzed their own cancer patient cohorts treated with different chemotherapy agents. They show that low SAMHD1 expression is favorable for the prognosis in three cancer types and that low expression is a predictive factor in lung and ovarian cancer treated with platin derivates and/or antimetabolites. Additionally, the authors underlined their observations with in vitro models to evaluate DNA damage and apoptosis after treatment with platinum derivatives. They generated SAMHD1 KO in a breast cancer cell line and ovary adenocarcinoma cell line and measured the expression of the DNA damage marker γH2AX and apoptosis by cleaved PARP expression.

The value of the study is reflected in the fact that (i) patient data and tumor material has been used and (ii) that the authors analyzed solid cancers, non-small cell lung cancer (NSCLC), ovarian and breast cancer cases, so far not yet studied. However, the in vitro studies need improvements to strengthen their conclusions.

Major points:

  1. γH2AX was measured by analyzing MFI in FACS. The gold standard for analyzing γH2AX are foci number in IF. As the differences of the measured MFI in the T47D cells are not great (4A) and not statistically significant, it is advised to use a different method to support the data.
  2. Fig 4D and suppl Fig 6D: Differences in expression of apoptosis marker cleaved PARP are not statistically significant and hardly visible. The authors should use an additional alternative marker to conclude on the data. Additionally, the statement in the result section (increased apoptosis upon treatment with platinum derivatives upon KD/KO of SAMDH1) needs to be adjusted to reflect the actual results.
  3. Suppl Figure 5 on cell viability contradicts the results on apoptosis. If there is more measurable apoptosis (shown by PARP cleavage), the KO cells should be more sensitive to treatment?
  4. It is interesting, that the authors observed SAMDH1 expression as a negative prognostic factor in breast and NSCLC patients in their cohorts. The authors should discuss their findings with respect to published data derived from TCGA or ICGC database (PMID: 33857133 and PMID: 34480199), where particularly in breast and lung cancers, high expression of SAMHD1 seems to be associated with survival
  5. Tone down the statement in the conclusions as dNTPase activity has not been measured, so no conclusion can be made regarding dNTPase activity “…which may be driven by SAMHD1 role in DNA damage function and not directly linked to its canonical dNTPase activity”
  1. Kaplan-Meier curves: black and blue lines are hardly distinguishable. Please use different colors
  2. Figure 4A is missing

Minor points:

1) some statements need corrections:

line 25: “Moreover, SAMHD1 expression is also a positive predictive factor…”: ..low SAMHD1 expression..

2) References are missing/incorrect formatting: please correct and check text throughout

Line 92: DNA damage sites4

Lines 271-272: delete journal information

Line 410: following DNA damage induction 26

Line 412: levels of H2AX,

Line 471: solid tumors as previously suggested31

Line 494: risk of metastases12,

Line 542: promoting cell survival49